# LEARNING TO PLAY ATARI IN A WORLD OF TOKENS

## ABSTRACT

Model-based reinforcement learning agents utilizing transformers have shown improved sample efficiency due to their ability to model extended context, resulting in more accurate world models. However, for complex reasoning and planning tasks, these methods primarily rely on continuous representations. This complicates modeling of discrete properties of the real world such as disjoint object classes between which interpolation is not plausible. In this work, we introduce discrete abstract representations for transformer-based learning (DART), a sample-efficient method utilizing discrete representations for modeling both the world and learning behavior. We incorporate a transformer-decoder for autoregressive world modeling and a transformer-encoder for learning behavior by attending to task-relevant cues in the discrete representation of the world model. For handling partial observability, we aggregate information from past time steps as memory tokens. DART outperforms previous state-of-the-art methods that do not use look-ahead search on the Atari 100k sample efficiency benchmark with a median human-normalized score of 0.790 and beats humans in 9 out of 26 games.

## 1 INTRODUCTION

A reinforcement learning (RL) algorithm usually takes millions of trajectories to master a task, and the training can take days or even months, especially when using complex simulators. This is where model-based reinforcement learning (MBRL) comes in handy (Sutton, 1991). With MBRL, agent learns the *dynamics* of the environment, understanding how environment state changes when different actions are taken. This method is more efficient because the agent can train in its *imagination* without requiring millions of trajectories (Ha & Schmidhuber, 2018). Additionally, the learned model allows the agent for safe and accurate decision-making by utilising different look-ahead search algorithms for planning its action (Hamrick et al., 2020).

Most MBRL methods commonly follow a structured three-step approach: 1) Representation Learning $\phi : S \rightarrow \mathbb{R}^n$, the agents captures a simplified representation $\mathbb{R}^n$ of the high dimensional environment state $S$; 2) Dynamics Learning $P : S \times A \times S \rightarrow [0, 1]$, the agent grasps how the environment evolves $P(s'|s,a)$ in response to its actions; and 3) Policy Learning $\pi : S \rightarrow \mathcal{P}(A)$, the agent determines the optimal actions needed to achieve its goals. Dreamer is a family of MBRL agents that follow a similar structured three-step approach.

DreamerV1 (Hafner et al., 2019) employed a recurrent state space model (RSSM) to learn the world model. DreamerV2 (Hafner et al., 2020), an improved version of DreamerV1, offers better sample efficiency and scalability by incorporating a discrete latent space for representing the environment. Building on the advancements of DreamerV2, DreamerV3 (Hafner et al., 2023) takes a similar approach with additions involving the use of symlog predictions and various regularisation techniques aimed at stabilizing learning across diverse environments. Notably, DreamerV3 surpasses the performance of past models across a wide range of tasks, while using fixed hyperparameters.

Although Dreamer variants are among the most popular MBRL approaches, they suffer from sample-inefficiency. The training of Dreamer models can require an impractical amount of gameplay time, ranging from months to thousands of years, depending on the complexity of the game. This inefficiency can be primarily attributed to inaccuracies in the learned world model, which tend to propagate errors into the policy learning process, resulting in compounding error problem. This challenge is largely associated with the use of convolutional neural networks (CNNs) and recur-

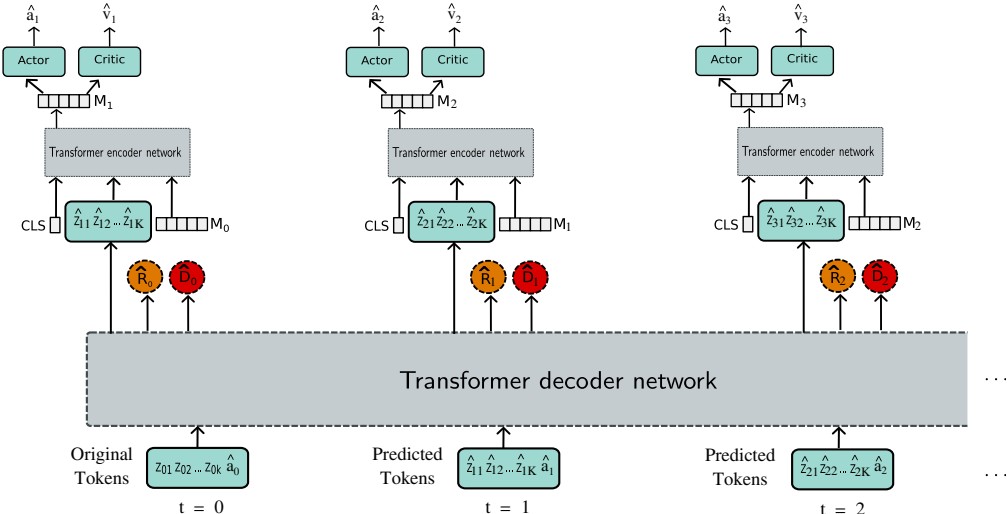

Figure 1: Discrete abstract representation for transformer-based learning (DART): In this approach, the original observation $x_t$ is encoded into discrete tokens $z_t$ using VQ-VAE. These tokenized observations, and predicted action, serve as inputs for modeling the world. Transformer decoder network is used for modeling the world. The predicted tokens, along with a `CLS` and a `MEM` token are used as input by the policy. This policy is modeled using a transformer-encoder network. The `CLS` token aggregates information from the observation tokens and the `MEM` token to learn a common representation, which is then used for action and value predictions. This common representation also plays a role in modeling memory, acting as the `MEM` token at the subsequent time step.

rent neural networks (RNNs) that, while effective in many domains, face limitations in capturing complex and long-range dependencies, which are common in RL scenarios.

This motivates the need to use transformers (Vaswani et al., 2017; Lin et al., 2022), which have proven highly effective in capturing long-range dependencies in various natural language processing (NLP) tasks (Wolf et al., 2020) and addressing complex visual reasoning challenges in computer vision (CV) tasks (Khan et al., 2022). Considering these advantages, recent works have adapted transformers for modeling the dynamics in MBRL. Transdreamer (Chen et al., 2022) first used a transformer-based world model by replacing Dreamer's RNN-based stochastic world model with a transformer-based state space model. It outperformed DreamerV2 in Hidden Order Discovery Tasks which requires long-term dependency and complex-reasoning. In order to stabilize the training, it utilizes gated transformer-XL (GTrXL) (Parisotto et al., 2020) architecture.

Masked world model (MWM) (Seo et al., 2023) utilizes a convolutional-autoencoder and vision transformer (ViT) (Dosovitskiy et al., 2020) for learning a representation that models dynamics following the RSSM objective. Their decoupling approach outperforms DreamerV2 on different robotic manipulation tasks from Meta-world (Yu et al., 2020) and RLBench (James et al., 2020). Similarly, transformer-based world model (TWM) (Robine et al., 2023a) use transformer-XL (TrXL) (Dai et al., 2019) for modeling the world and use the predicted latent states for policy learning. Their work demonstrates sample-efficient performance on the Atari 100k benchmark.

Contrary to these approaches, imagination with auto-regression over an inner speech (IRIS) (Micheli et al., 2022) models dynamics learning as a sequence modeling problem, utilizing discrete image tokens for modeling the world. It then uses reconstructed images using the predicted tokens for learning the policy using CNNs and long short-term memorys (LSTMs), achieving improved sample efficiency on the Atari 100k compared to past models. However, it still faces difficulties in modeling the memory leading to the problem of partial observability, resulting in reduced performance.

In this work, we introduce discrete abstract representation for transformer-based learning (DART), a novel approach that leverages transformers for learning both the world model and policy. Unlike the previous method Yoon et al. (2023), which solely utilized a transformer for extracting object-centric representation, our approach employs a transformer to learn behavior through discrete representa-

tion (Mao et al., 2021), as predicted by the transformer-decoder that models the world. This choice allows the model to focus on fine-grained details, facilitating precise decision-making. Specifically, we utilize a transformer-decoder architecture, akin to the generative pre-trained transformer (GPT) framework, to model the world, while adopting a transformer encoder, similar to the ViT architecture, to learn the policy (as illustrated in Figure 1).

Additionally, to address challenges related to partial observability necessitates memory modeling. Previous work Didolkar et al. (2022) modeled memory in transformers using a computationally intensive two-stream network. Inspired by (Bulatov et al., 2022), we model memory as a distinct token, aggregating task-relevant information over time using a self-attention mechanism.

**Contributions** We outline the significant contribution of our work as follows:

- **Transformer-Based Modeling**: Our work introduces a novel approach that utilizes transformers for both world and policy modeling. Specifically, we utilize a transformer-decoder (GPT) for world modeling and a transformer-encoder (ViT) for policy learning. This represents an improvement compared to IRIS, which relies on CNNs and LSTMs for policy learning, potentially limiting its performance.
- **Utilizing Discrete Representations**: We use discrete representations for policy and world modeling. These discrete representations capture abstract features, enabling our transformer-based model to focus on task-specific fine-grained details. Attending to these details improves decision-making, as demonstrated by our results.
- **Efficient Handling of Partial Observability**: To address the problem of partial observability, we introduce a novel mechanism for modeling the memory that aggregates task-relevant information from the previous time step to the next using a self-attention mechanism.
- **Enhanced Interpretability and Sample Efficiency**: Our model showcases enhanced interpretability and sample efficiency. It achieves state-of-the-art results (no-look-ahead search methods) on the Atari 100k benchmark with a median score of 0.790 and superhuman performance in 9 out of 26 games.

## 2 METHOD

Our model, DART, is designed for mastering Atari games, within the framework of a partially observable Markov decision process (POMDP) (Kaelbling et al., 1998) which is defined as a tuple $(\mathcal{O}, \mathcal{A}, p, r, \gamma, d)$. Here, $\mathcal{O}$ is the observation space with image observations $x_t \subseteq \mathbb{R}^{h \times w \times 3}$, $\mathcal{A}$ represents the action space, and $a_t$ is a discrete action taken at time step $t$ from the action space $\mathcal{A}$, $p(x_t \mid x_{<t}, a_{<t})$ is the transition dynamics, r is the reward function $r_t = r(x_{\leq t}, a_{<t})$, $\gamma \in [0, 1)$ is the discount factor and $d \in \{0, 1\}$ indicates episode termination. The goal is to find a policy $\pi$ that maximizes the expected sum of discounted rewards $\mathbb{E}_\pi \left[ \sum_{t=1}^{\infty} \gamma^{t-1} r_t \right]$. DART comprises three main steps: (1) *Representation Learning*, where vector quantised-variational autoencoders (VQ-VAEs) (Van Den Oord et al., 2017) are used for tokenizing the original observations; (2) *World-Model Learning*, which involves auto-regressive modeling of the dynamics of the environment using GPT architecture; and (3) *Policy Learning*, which is modeled using ViT for decision-making by attending to task-relevant cues. We now describe our overall approach in detail.

### 2.1 REPRESENTATION LEARNING

Discrete symbols are essential in human communication, as seen in natural languages (Cartuyvels et al., 2021). Likewise, in the context of RL, discrete representation is useful for abstraction and reasoning, leveraging the inherent structure of human communication (Islam et al., 2022). This motivates our approach to model the observation space as a discrete set. In this work, we use VQ-VAE for discretizing the observation space. It learns a discrete latent representation of the input data by quantizing the continuous latent space into a finite number of discrete codes.

$$
\begin{aligned}
&\text{Image Encoder: } \hat{z}_t^k = f_\theta(x_t), &&\text{Vector Quantization: } \hat{z}_q, \mathcal{L}_{\text{VQ}} = q(\hat{z}_t^k; \phi_q, Z), \\
&\text{Codebook: } \quad Z = \{z_1, z_2, \ldots, z_N\}, &&\text{Image Decoder: } \quad \hat{x}_t = g_\phi(\hat{z}_{q_t}^k).
\end{aligned}
\tag{1}
$$

At time step $t$, the observation from the environment $x_t \in \mathbb{R}^{H \times W \times 3}$, is encoded by the image encoder $f_\theta$ to a continuous latent space $\hat{z}_t^k$. This encoder is modeled using CNNs. The quantization

process $q$ maps the predicted continuous latent space $\hat{z}_t^k$ to a discrete latent space $\hat{z}_q$. This is done by finding the closest embedding vector in the codebook from a set of $N$ codes. The discrete latent codes are passed to the decoder $g_\phi$, which maps it back to the input data $\hat{x}_t$ (see Equation 1).

The training of this VQ-VAE comprises minimizing the *reconstruction loss* to ensure alignment between input and reconstructed images. Simultaneously, the codebook is learned by minimizing the *codebook loss*, encouraging the embedding vector in the codebook to be close to the encoder output. The *commitment loss* encourages the encoder output to be close to the nearest codebook vector. Additionally *perceptual loss* is computed to encourage the encoder to capture high-level features. The total loss in VQ-VAE is a weighted sum of these loss functions.

This approach enables the modeling of fine-grained, low-level information within the input image as a set of discrete latent codes.

## 2.2 WORLD-MODEL LEARNING

The discrete latent representation forms the core of our approach, enabling the learning of dynamics through an autoregressive next-token prediction approach (see Equation 2). A transformer decoder based on the GPT architecture is used for modeling this sequence prediction framework:

$$
\begin{aligned}
&\text{Aggregate Sequence: } \hat{z}_{ct} = f_\phi(\hat{z}_{<t}, \hat{a}_{<t}), &&\text{Next State Token Predictor: } \hat{z}_{q_t}^k \sim p_d(\hat{z}_{q_t}^k \mid \hat{z}_{ct}), \\
&\text{Reward Predictor: } \quad \hat{r}_t \sim p_d(\hat{r}_t \mid \hat{z}_{ct}), &&\text{Episode End Predictor: } \quad \hat{d}_t \sim p_d(\hat{d}_t \mid \hat{z}_{ct}).
\end{aligned}
\tag{2}
$$

First, an aggregate sequence is modeled by encoding past latent tokens and actions at each time step. The aggregated sequence is used for estimating the distribution of the next token, contributing to the modeling of future states. Simultaneously, it is also used for estimating the reward and the episode termination. This training occurs in a self-supervised manner, with the next state predictor and termination modules trained using cross-entropy loss, while reward prediction uses mean squared error.

## 2.3 POLICY-LEARNING

The policy $\pi$ is trained within the world model, using a transformer encoder architecture based on vision transformer (ViT). At each time step $t$, the policy processes the current observation as $K$ discrete tokens received from the world model. These observation tokens are extended with additional learnable embeddings, including a `CLS` token placed at the beginning and a `MEM` token appended to the end.

The `CLS` token helps in aggregating information from the $K$ observation tokens and the `MEM` token. Meanwhile, the `MEM` token acts as a memory unit, accumulating information from the previous time steps. Thus, at time step $t$ the input to the policy can be represented as $(\texttt{CLS}, \hat{z}_{q_t}^1, \ldots, \hat{z}_{q_t}^K, \texttt{MEM}_t)$, where $\hat{z}_{q_t}^K$ corresponds to the embedding of $K^{th}$ index token from the codebook, such that

$$
\begin{aligned}
&\mathbf{out} = [\texttt{CLS}, \hat{z}_{q_t}^1, \ldots, \hat{z}_{q_t}^K, \texttt{MEM}_{t-1}] + \mathbf{E}_{\text{pos}}, &&\texttt{CLS}, \hat{z}_{q_t}, \texttt{MEM}_{t-1}, \mathbf{E}_{\text{pos}} \in \mathbb{R}^D \\
&\left.\begin{aligned}
&\mathbf{out} = \mathbf{out} + \text{MSA}(\text{LN}(\mathbf{out})), \\
&\mathbf{out} = \mathbf{out} + \text{MLP}(\text{LN}(\mathbf{out})),
\end{aligned}\right\} \times \text{L} \\
&\mathbf{h}_t = \mathbf{out}[0], \quad \texttt{MEM}_t = \mathbf{out}[0].
\end{aligned}
\tag{3}
$$

While these discrete tokens excel at capturing fine-grained low-level details, they lack spatial information about various features or objects within the image. Transformers, known for their permutational-equivariant nature, efficiently model global representation. To incorporate local spatial information, we add learnable positional encoding to the original input. During training, these embeddings converge into vector spaces that represent the spatial location of different tokens.

Following this spatial encoding step, the output is first processed with layer-normalization within the residual block. This helps in enhancing gradient flow and eliminates the need for an additional warm-up strategy as recommended in Xiong et al. (2020). Subsequently, the output undergoes processing via multi-head self-attention (MSA) and a multi-layer perception (see Equation 3). This series of operations is repeated for a total of $L$ blocks.

$$\text{Actor:} \quad \hat{a}_t \sim p_\psi \left( \hat{a}_t \mid \hat{h}_t \right), \quad \text{Critic:} \quad v_\xi \left( \hat{h}_t \right) \approx \mathbb{E} p_\psi \left[ \sum_{\tau \geq t} \hat{\gamma}^{\tau - t} \hat{r}_\tau \right]. \tag{4}$$

Following $L$ blocks of operations, the feature vector associated with the `CLS` token serves as the representation, modeling both the current state and memory. This representation $\mathbf{h_t}$, is used by the policy to sample action and by the critic to estimate the expected return (see Equation 4). This is followed by the reward prediction, episode end prediction, and the token predictions of the next observation by the world model.

The feature vector $\mathbf{h_t}$ now becomes the memory unit. This is possible because the self-attention mechanism acts like a gate, passing on information to the next time step as required by the task. This simple approach enables effective memory modeling without relying on recurrent networks, which can be challenging to train and struggle with long context (Pascanu et al., 2013).

The imagination process unfolds for a duration of $H$ steps, stopping on episode-end prediction. Similar to the IRIS and DreamerV2 approaches, we optimize the policy by minimizing $\mathcal{L}_V$ and $\mathcal{L}_\pi$, defined as follows:

$$V_t^\lambda = \begin{cases} \hat{r}_t + \gamma \left( 1 - \hat{d}_t \right) \left[ (1 - \lambda) v \left( \hat{x}_{t+1} \right) + \lambda V_{t+1}^\lambda \right] & \text{if } t < H \\ v \left( \hat{x}_H \right) & \text{if } t = H \end{cases},$$

$$\mathcal{L}_V = \mathbb{E}_\pi \left[ \sum_{t=0}^{H-1} \left( V \left( \hat{x}_t \right) - \text{sg} \left( V_t^\lambda \right) \right)^2 \right], \tag{5}$$

$$\mathcal{L}_\pi = -\mathbb{E}_\pi \left[ \sum_{t=0}^{H-1} \log \left( \pi \left( a_t \mid \hat{x}_{\leq t} \right) \right) \text{sg} \left( V_t^\lambda - V \left( \hat{x}_t \right) \right) + \eta \mathcal{H} \left( \pi \left( a_t \mid \hat{x}_{\leq t} \right) \right) \right].$$

## 3 EXPERIMENTS

We evaluated our model alongside existing baselines using the Atari 100k benchmark (Kaiser et al., 2019), a commonly used testbed for assessing the sample-efficiency of RL algorithms. It consists of 26 games from the Arcade Learning Environment (Bellemare et al., 2013), each with distinct settings requiring perception, planning, and control skills.

We evaluated our model's performance based on several metrics, including the mean and median of the human-normalized score, which measures how well the agent performs compared to human and random players given as $\frac{\text{score}_{\text{agent}} - \text{score}_{\text{random}}}{\text{score}_{\text{human}} - \text{score}_{\text{random}}}$. We also used the super-human score to quantify the number of games in which our model outperformed human players. We further evaluated our model's performance using the Interquartile Mean (IQM) score and the Optimality Gap, following the evaluation guidelines outlined in (Agarwal et al., 2021)

We rely on the median score to evaluate overall model performance, as it is less affected by outliers. The mean score can be strongly influenced by a few games with exceptional or poor performance. Additionally, the IQM score helps in assessing both consistency and average performance across all games.

Atari environments offer the model an RGB observation of $64 \times 64$ dimensions, featuring a discrete action space, and the model is allowed to be trained using only 100k environment steps (equivalent to 400k frames due to a frameskip of 4), which translates to approximately 2 hours of real-time gameplay.

### 3.1 RESULTS

In Figure 2, we present the IQM and optimality gap scores, as well as the mean and median scores. These scores pertain to various models assessed on Atari 100k. Figure 3a visualizes the performance profile, while Figure 3b illustrates the probability of improvement, which quantifies the likelihood of DART surpassing baseline models in any Atari game. To perform these comparisons, we use results from Micheli et al. (2022), which include scores of 100 runs of CURL (Laskin et al., 2020), DrQ (Kostrikov et al., 2020), SPR (Schwarzer et al., 2020), as well as data from 5 runs of Sim-PLe (Kaiser et al., 2019), and IRIS.

Table 1: DART achieves a new state-of-art median score among no-look-ahead search methods. It attains the highest median score, interquartile mean (IQM), and optimality gap score. Moreover, DART outperforms humans in 9 out of 26 games and achieves a higher score than IRIS in 18 out of 26 games (underlined).

| | | | | | No look-ahead search | | |
| | | | | | | Transformer based | |
| Game | Random | Human | SPR | DreamerV3 | TWM | IRIS | DART |
|------|--------|-------|-----|-----------|-----|------|------|
| Alien | 227.8 | 7127.7 | 841.9 | 959 | 674.6 | 420.0 | **962.0** |
| Amidar | 5.8 | 1719.5 | **179.7** | 139 | 121.8 | 143.0 | 125.7 |
| Assault | 222.4 | 742.0 | 565.6 | 706 | 682.6 | **1524.4** | 1316.0 |
| Asterix | 210.0 | 8503.3 | 962.5 | 932 | **1116.6** | 853.6 | 956.2 |
| BankHeist | 14.2 | 753.1 | 345.4 | **649** | 466.7 | 53.1 | 629.7 |
| BattleZone | 2360.0 | 37187.5 | 14834.1 | 12250 | 5068.0 | 13074.0 | **15325.0** |
| Boxing | 0.1 | 12.1 | 35.7 | 78 | 77.5 | 70.1 | **83.0** |
| Breakout | 1.7 | 30.5 | 19.6 | 31 | 20.0 | **83.7** | 41.9 |
| ChopperCommand | 811.0 | 7387.8 | 946.3 | 420 | **1697.4** | 1565.0 | 1263.8 |
| CrazyClimber | 10780.5 | 35829.4 | 36700.5 | **97190** | 71820.4 | 59324.2 | 34070.6 |
| DemonAttack | 152.1 | 1971.0 | 517.6 | 303 | 350.2 | 2034.4 | **2452.3** |
| Freeway | 0.0 | 29.6 | 19.3 | 0 | 24.3 | 31.1 | **32.2** |
| Frostbite | 65.2 | 4334.7 | 1170.7 | 909 | **1475.6** | 259.1 | 346.8 |
| Gopher | 257.6 | 2412.5 | 660.6 | **3730** | 1674.8 | 2236.1 | 1980.5 |
| Hero | 1027.0 | 30826.4 | 5858.6 | **11161** | 7254.0 | 7037.4 | 4927.0 |
| Jamesbond | 29.0 | 302.8 | 366.5 | 445 | 362.4 | **462.7** | 353.1 |
| Kangaroo | 52.0 | 3035.0 | 3617.4 | **4098** | 1240.0 | 838.2 | 2380.0 |
| Krull | 1598.0 | 2665.5 | 3681.6 | **7782** | 6349.2 | 6616.4 | 7658.3 |
| KungFuMaster | 258.5 | 22736.3 | 14783.2 | 21420 | **24554.6** | 21759.8 | 23744.3 |
| MsPacman | 307.3 | 6951.6 | 1318.4 | 1327 | **1588.4** | 999.1 | 1132.7 |
| Pong | -20.7 | 14.6 | -5.4 | 18 | **18.8** | 14.6 | 17.2 |
| PrivateEye | 24.9 | 69571.3 | 86.0 | **882** | 86.6 | 100.0 | 765.7 |
| Qbert | 163.9 | 13455.0 | 866.3 | **3405** | 3330.8 | 745.7 | 750.9 |
| RoadRunner | 11.5 | 7845.0 | 12213.1 | **15565** | 9109.0 | 4046.2 | 7772.5 |
| Seaquest | 68.4 | 42054.7 | 558.1 | 618 | 774.4 | 661.3 | **895.8** |
| UpNDown | 533.4 | 11693.2 | 10859.2 | 7667 | **15981.7** | 3546.2 | 3954.5 |
| #Superhuman(↑) | 0 | N/A | 6 | 9 | 7 | 9 | 9 |
| Mean(↑) | 0.000 | 1.000 | 0.616 | 1.120 | 0.956 | 1.046 | 1.022 |
| Median(↑) | 0.000 | 1.000 | 0.396 | 0.466 | 0.505 | 0.289 | **0.790** |
| IQM(↑) | 0.000 | 1.000 | 0.337 | 0.490 | - | 0.501 | **0.575** |
| Optimality Gap(↓) | 1.000 | 0.000 | 0.577 | 0.508 | - | 0.512 | **0.458** |

DART exhibits a similar mean performance as IRIS. However, the median and IQM scores show that DART outperforms other models consistently.

Table 1 presents DART's score across all 26 games featured in the Atari 100k benchmark. We compare its performance against other strong world models including DreamerV3 (Hafner et al., 2023), as well as other transformer-based world models such as TWM (Robine et al., 2023a) and IRIS (Micheli et al., 2022).

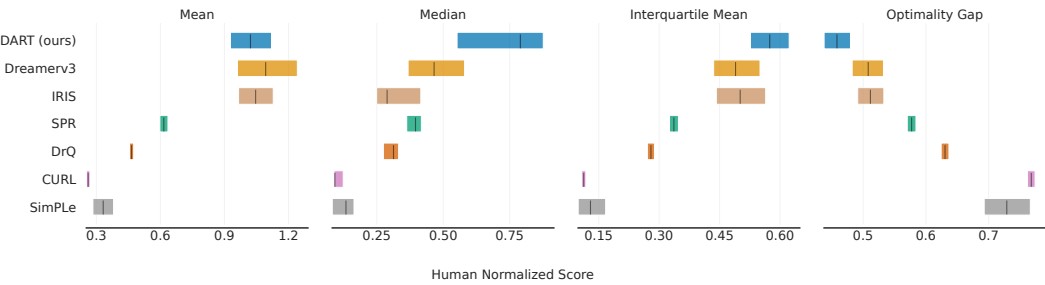

Figure 2: Comparison of Mean, Median, and Interquartile Mean Human-Normalized Scores

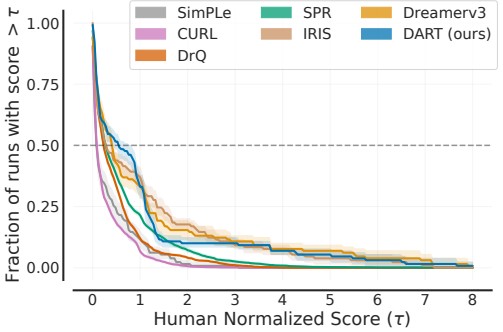
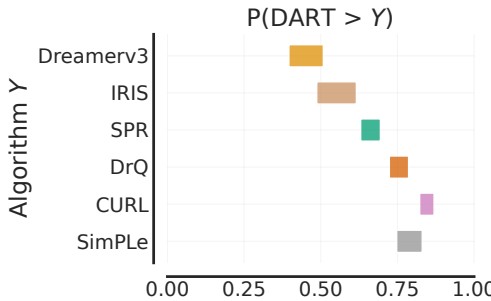

(a) The performance profiles on the Atari 100k benchmark illustrate the proportion of runs across all games (y-axis) that achieve a score normalized against human performance (x-axis).

(b) The probabilities of improvement visualized here refer to the likelihood of DART surpassing the performance of baseline models in any game.

Figure 3: Comparison of different models using performance profiles and probabilities of improvement.

To assess DART's overall performance, we calculate the average score over 100 episodes post-training, utilizing five different seeds to ensure robustness. DART outperforms the previous best model, IRIS, in 18 out of 26 games. It achieves a median score of 0.790 (an improvement of 61% when compared to DreamerV3). Additionally, it reaches an IQM of 0.575 reflecting a 15% advancement, and significantly improves the OG score to 0.458, indicating a 10% improvement when compared to IRIS. DART also achieves a superhuman score of 9, outperforming humans in 9 out of 26 games.

## 3.2 POLICY ANALYSIS

In Figure 4, we present the attention maps for the 6 layers of our transformer policy using a heatmap visualization. These maps are generated by averaging the attention scores from each multi-head attention mechanism across all layers. The final visualization is obtained by further averaging these attention maps over 20 randomly selected observation states during an episode. This analysis provides insights into our approach to information aggregation through self-attention.

The visualization in Figure 4 shows that the extent to which information is aggregated from the past and the current state to the next state depends on the specific task at hand. In games featuring slowly moving objects where the current observation provides complete information to the agent, the memory token receives less attention (see Figure 4a). Conversely, in environments with fast-moving objects like balls and paddles, where the agent needs to model the past trajectory of objects (e.g., Breakout and Private Eye), the memory token is given more attention (see Figure 4b- 4d). This observation highlights the adaptability of our approach to varying task requirements.

Table 2: Evaluating DART's performance through various techniques such as memory token masking, random observation masking, and the removal of positional encoding and random exploration.

| Game | Original | w/o | | Masked Memory | Masked Observation Token | | | |
| | | PE | $\epsilon$ | | 25% | 50% | 75% | 100% |
|---|---|---|---|---|---|---|---|---|
| Boxing | 83.0 | 3.86 | 58.67 | 81.45 | 77.79 | 51.14 | 15.64 | -11.91 |
| Amidar | 125.7 | 77.1 | 92.75 | 113.69 | 102.47 | 56.37 | 52.22 | 30.43 |
| Road Runner | 7772.5 | 1030.0 | 3597.1 | 8021.0 | 7354.0 | 2730.0 | 988.0 | 961.0 |
| Seaquest | 895.8 | 64.2 | 753.93 | 704.8 | 491.4 | 207.8 | 104.0 | 142.0 |
| KungFuMaster | 23744.3 | 1028.0 | 15464.7 | 20378.0 | 16436.0 | 9760.0 | 4676.2 | 1571.8 |

## 3.3 ABLATION STUDIES

We further analyzed DARTs performance across various experimental settings, as detailed in Table 2 for five distinct games. The original score of DART is presented in the second column. The different scenarios include:

**Without Positional Encoding (PE):** The third column demonstrates the performance of DART when learned positional encoding is excluded. We can observe that in environments where agents need to closely interact with their surroundings, such as in Boxing and KungFuMaster, the omission of positional encoding significantly impacts performance. However, in games where the enemy may not be in close proximity to the agent, such as Amidar, there is a slight drop in performance without positional encoding. This is because transformers inherently model global context, allowing the agent to plan its actions based on knowledge of the overall environment state. However, precise decision-making requires positional information about the local context. In our case, adding learnable positional encoding provides this, resulting in a significant performance boost.

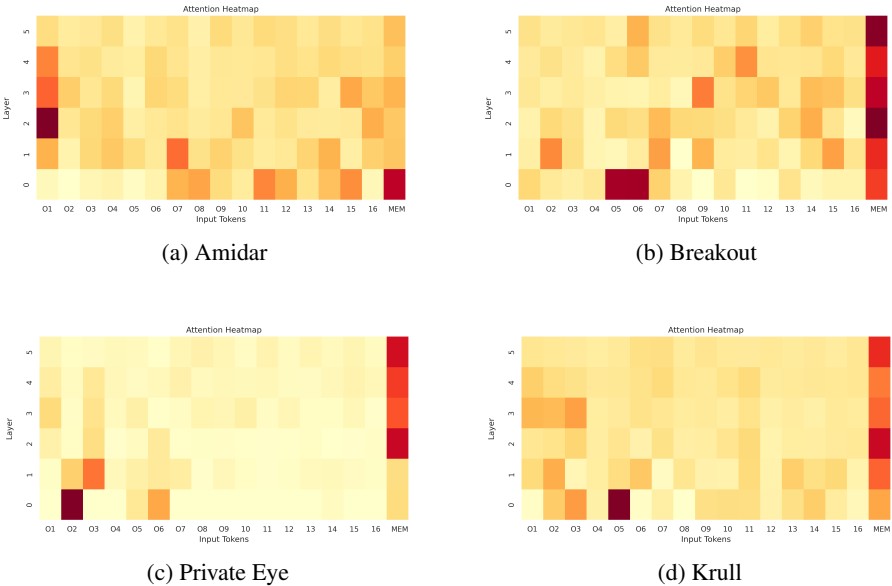

(a) Amidar

(b) Breakout

(c) Private Eye

(d) Krull

Figure 4: **Comparison of Memory Requirements Across Atari Games:** Atari games exhibit varying memory requirements, depending on their specific dynamics. Games with relatively static or slow-moving objects, like *Amidar*, maintain complete information at each time step and thus aggregate less information from the memory token. Conversely, games characterized by rapidly changing environments, such as *Breakout*, *Krull*, and *PrivateEye*, require modeling the past trajectories of objects. As a result, the policy for these games heavily relies on the memory token to aggregate information from past states into future states.

**No Exploration ($\epsilon$):** The fourth column illustrates DARTs performance when trained without random exploration, relying solely on agent-predicted actions for collecting trajectories for world modeling. However, like IRIS, our model also faces the double-exploration challenge. This means that the agent's performance declines when new environment states aren't introduced through random exploration, which is crucial for effectively modeling the dynamics of the world. It's worth noting that for environments with simpler dynamics (e.g., Seaquest), the performance impact isn't as substantial.

**Masking Memory Tokens:** In the fifth column, we explore the impact of masking the memory token, thereby removing past information. Proper modeling of memory is crucial in RL to address the challenge of partial observability and provide information about various states (e.g., the approaching trajectory of a ball, and the velocity of the surrounding objects) that are important for decision-making. Our method of aggregating memory over time enhances DARTs overall performance. It is interesting to observe improvement in the agent's performance with masked memory

tokens in the case of RoadRunner. This could be because the original state already contains complete information, rendering the memory token redundant, thereby impacting the final performance.

**Random Observation Token Masking:** The last set of columns explores the consequences of randomly masking observation tokens, which selectively removes low-level information. Given that each token among the $K$ tokens model distinct low-level features of the observation, random masking has a noticeable impact on the agent's final performance. When observation tokens are masked 100%, the agent attends solely to the memory token, resulting in a significant drop in overall performance.

## 4 RELATED WORK

**Sample Efficiency in RL.** Enhancing sample efficiency (i.e., the amount of data required to reach a specific performance level) constitutes a fundamental challenge in the field of RL. This efficiency directly impacts the time and resources needed for training an RL agent. Numerous approaches aimed at accelerating the learning process of RL agents have been proposed (Buckman et al., 2018; Mai et al., 2022; Yu, 2018). Model-based RL is one such approach that helps improve the sample efficiency. It reduces the number of interactions an agent needs to have with the environment to learn the policy (Moerland et al., 2023; Polydoros & Nalpantidis, 2017; Atkeson & Santamaria, 1997). This is done by allowing the policy to learn the task in the imagined world (Wang et al., 2021b; Mu et al., 2021; Okada & Taniguchi, 2021; Zhu et al., 2020), motivating the need to have an accurate world model while providing the agent with concise and meaningful task-relevant information for faster learning. Considering this challenge (Kurutach et al., 2018) learns an ensemble of models to reduce the impact of model bias and variance. Uncertainty estimation is another approach as shown in (Plaat et al., 2023) to improve model accuracy. It involves estimating the uncertainty in the model's prediction so that the agent focuses its exploration in those areas. The other most common approach for an accurate world model is using a complex or higher-capacity model architecture that is better suited to the task at hand (Wang et al., 2021a; Ji et al., 2022). For example, using a transformer-based world model, as in TransDreamer (Chen et al., 2022), TWM (Robine et al., 2023a), and IRIS (Micheli et al., 2022).

Learning a low-dimensional representation of the environment can also help improve the sample efficiency of RL agents. By reducing the dimensionality of the state, the agent can learn an accurate policy with fewer interactions with the environment (McInroe et al., 2021; Du et al., 2019). Variational Autoencoders (VAEs) (Kingma et al., 2019) are commonly used for learning low-dimensional representations in MBRL (Andersen et al., 2018). The VAEs capture a compact and informative representation of the input data. This allows the agent to learn the policy faster (Ke et al., 2018; Corneil et al., 2018). However, VAEs learn a continuous representation of the input data by forcing the latent variable to be normally distributed. This poses a challenge for RL agents, where agents need to focus on precise details for decision-making (Dunion et al., 2022).(Lee et al., 2020) show disentangling representations helps in modeling interpretable policy and improves the learning speed of RL agents on various manipulation tasks. Recent works (Robine et al., 2023b; Zhang et al., 2022) have used VQ-VAE for learning independent latent representations of different low-level features present in the original observation. Their clustering properties have enabled robust, interpretable, and generalizable policy across a wide range of tasks.

## 5 DISCUSSION

**Conclusion** In this work, we introduced DART, a model-based reinforcement learning agent that learns both the model and the policy using discrete tokens. Through our experiments, we demonstrated our approach helps in improving performance and achieves a new state-of-the-art score on the Atari 100k benchmarks for methods with no look-ahead search during inference. Moreover, our approach for memory modeling and the use of a transformer for policy modeling provide additional benefits in terms of interpretability.

**Limitation** As of now, our method is primarily designed for environments with discrete action spaces. This limitation poses a significant challenge, considering that many real-world robotic control tasks necessitate continuous action spaces. For future work, it would be interesting to adapt our approach to continuous action spaces and modeling better-disentangled tokens for faster learning.

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

# A APPENDIX

## A.1 EXPERIMENT ON CRAFTER

Crafter (Hafner, 2021), inspired by Minecraft (Guss et al., 2019), allows assessing an agent's general abilities within a single environment. This distinguishes it from Atari 100k, where the agent must be evaluated across 26 different games that test for different skills. In Crafter, 2D worlds are randomly generated, featuring diverse landscapes like forests, lakes, mountains, and caves on a 64×64 grid. Players aim to survive by searching for essentials like food, water, and shelter while defending against monsters, collecting materials, and crafting tools. This setup allows for evaluating a wide range of skills within a single environment, spanning multiple domains, and increasing assessment comprehensiveness. The environment is partially observable with observations covering a small 9×9 region centered around the agent.

| Model | DreamerV3 | IRIS | DART |
|---|---|---|---|
| Steps | 200K | 200K | 200K |
| Return | $5.02 \pm 0.03$ | $5.45 \pm 0.21$ | $\mathbf{6.13 \pm 0.09}$ |

Table 3: Comparing the sample efficiency of DreameV3, IRIS, and DART on challenging Crafter environment which involves long-horizon tasks. Reported returns are specified as average and standard deviation over 5 seeds.

In the preliminary results shown in Table 3, we compare DART with IRIS in low data regime and observed that DART achieves a higher average return.

## A.2 EXPERIMENT ON ATARI WITH MORE ENVIRONMENT STEPS

| **Environment** | **Steps** $(k)$ | **Score** |
|---|---|---|
| Freeway | 100k | $32.2 \pm 0.57$ |
| | 150k | $33.1 \pm 0.37$ |
| KungFuMaster | 100k | $23744.3 \pm 3271,53$ |
| | 150k | $24756.5 \pm 2635.21$ |
| Pong | 100k | $17.2 \pm 1.74$ |
| | 150k | $17.6 \pm 2.79$ |

Table 4: Performance of DART with 100k and 150k environment steps ($k$). All results are shown as average and standard deviation over 3 seeds.

By training it beyond 100k training steps, we see improved performance of DART as shown in Table 4.

## A.3 MODEL CONFIGURATION

Recent works have used transformer-based architectures for MBRL. In Table 5 we compare the configurations used by different approaches for representation learning, world modeling, and behavior learning.

| | MWM | TWM | IRIS | DreamerV3 | STORM | DART |
|---|---|---|---|---|---|---|
| Parameters | n/a | n/a | 3.04M | 18M | n/a | 3.07M |
| Agent state | Continuous | Continuous | Continuous | Continuous | Continuous | Discrete |
| State model | MLP | MLP | CNN | MLP | MLP | ViT |
| Agent memory | ViT | Tr-XL | LSTM | GRU | GPT | ViT (Self-attention) |
| Representation | MAE | Cat.-VAE | VQ-VAE | Cat.-VAE | Cat.-VAE | VQ-VAE |

Table 5: Comparing the model configuration of recent MBRL approaches. n/a- Not Available; Cat.-VAE - Categorical VAE.

## A.4 HYPERPARAMETERS

A detailed list of hyperparameters is provided for each module, Table 6 for Image Tokenizer, Table 7 for World Modeling, and Table 8 for behaviour learning.

Table 6: Hyperparameters for image tokenization using VQ-VAE.

| Hyperparameter | Symbol | Value |
|---|---|---|
| Encoder convolutional layers | – | 4 |
| Decoder convolutional layers | – | 4 |
| Per layer residual blocks | – | 2 |
| Self-attention layers | – | 8 / 16 |
| Codebook size | $N$ | 512 |
| Embedding dimension | $d$ | 512 |
| Input image resolution | – | 64×64 |
| Image channels | – | 3 |
| Activation | – | Swish |
| Tokens per image | $K$ | 16 |
| Batch size | – | 64 |
| Learning rate | – | 0.0001 |

Table 7: Hyperparameters used for modeling the dynamics using transformer decoder.

| Hyperparameter | Symbol | Value |
|---|---|---|
| Embedding dimension | – | 256 |
| Transformer layers | – | 10 |
| Attention heads | – | 4 |
| Imagination steps | $H$ | 20 |
| Embedding dropout | – | 0.1 |
| Weight decay | – | 0.01 |
| Attention dropout | – | 0.1 |
| Residual dropout | – | 0.1 |
| Attention type | – | Causal |
| Activation | – | GeLU |
| Batch size | – | 64 |
| Learning rate | – | 0.0001 |

Table 8: Hyperparameters used for modeling behavior using transformer encoder.

| Hyperparameter | Symbol | Value |
|---|---|---|
| Input tokens | – | 18 |
| Embedding dimension | – | 512 |
| Attention heads | – | 8 |
| Transformer layers | $L$ | 6 |
| Dropout | – | 0.2 |
| Activation | – | GeLU |
| Transformer layers | – | 6 |
| Attention type | – | Self-attention |
| Positional embedding | – | Learnable |
| Gamma | $\gamma$ | 0.995 |
| Lambda | $\lambda$ | 0.95 |
| Batch size | – | 64 |
| Epsilon | $\epsilon$ | 0.01 |
| Temperature (train) | – | 1.0 |
| Temperature (test) | – | 0.5 |
| Learning rate | – | 0.0001 |

## A.5 WORLD-MODEL ACCURACY

To evaluate the accuracy of the world model, it is important to assess both the next-state prediction accuracy and reward prediction accuracy. Illustrated in Fig. 5, are the imagined trajectories for the

games of Pong, Krull, and KungFuMaster. The graphs demonstrate the accuracy of the world model in predicting subsequent states and rewards as the training progresses. In Fig. 6, we extend our analysis of the world model's efficiency in handling long-horizon tasks for the challenging Crafter environment. Our analysis shows that the transformer-based world model is able to learn a better world model as demonstrated by the peak signal-to-noise ratio (PSNR) values in Table 9.

Table 9: Comparing the PSNR values for the imagined trajectories produced by the world models of DreamerV3 and DART. The final PSNR is calculated by averaging the values obtained from 100 episodes, with each episode comprising 200 steps.

|  | DreamerV3 | DART |
|---|---|---|
| PSNR | 31.94dB | 33.53dB |

Table 10: Comparing the performance of DART with STORM.

| Game | STORM | DART |
|---|---|---|
| Alien | 984 | 962.0 |
| Amidar | 205 | 125.7 |
| Assault | 801 | 1316.0 |
| Asterix | 1028 | 956.2 |
| BankHeist | 641 | 629.7 |
| BattleZone | 13540 | 15325.0 |
| Boxing | 80 | 83.0 |
| Breakout | 16 | 41.9 |
| ChopperCommand | 1888 | 1263.8 |
| CrazyClimber | 66776 | 34070.6 |
| DemonAttack | 165 | 2452.3 |
| Freeway | 0 | 32.2 |
| Frostbite | 1316 | 346.8 |
| Gopher | 8240 | 1980.5 |
| Hero | 11044 | 4927.0 |
| Jamesbond | 509 | 353.1 |
| Kangaroo | 4208 | 2380.0 |
| Krull | 8413 | 7658.3 |
| KungFuMaster | 26182 | 23744.3 |
| MsPacman | 2673 | 1132.7 |
| Pong | 11 | 17.2 |
| PrivateEye | 7781 | 765.7 |
| Qbert | 4522 | 750.9 |
| RoadRunner | 17564 | 7772.5 |
| Seaquest | 525 | 895.8 |
| UpNDown | 7985 | 3954.5 |
| #Superhuman($\uparrow$) | 9 | 9 |
| Mean($\uparrow$) | 1.267 | 1.022 |
| Median($\uparrow$) | 0.584 | 0.790 |

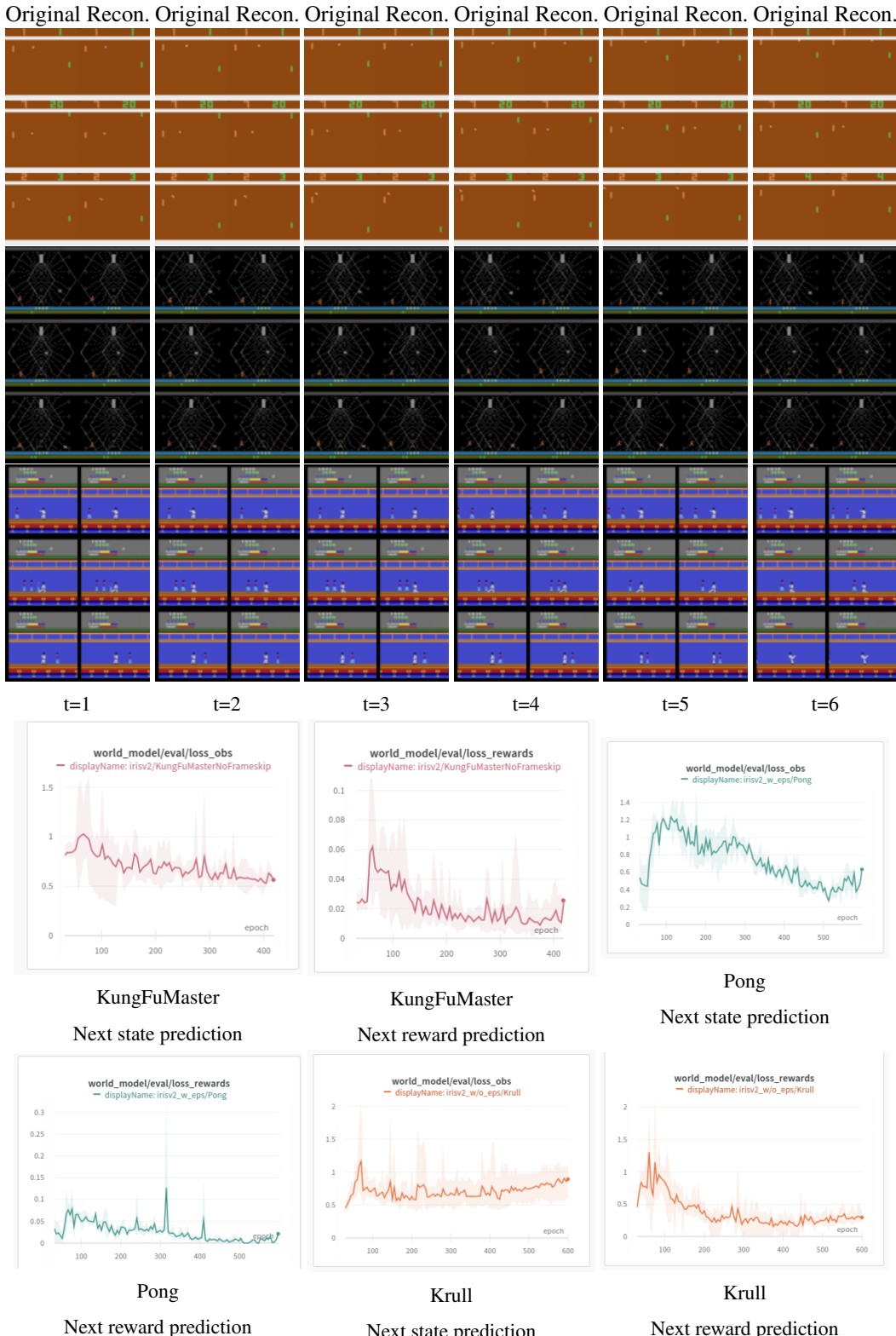

Figure 5: Analysing the accuracy of the transformer-based learned world model by visualizing the future imagined trajectories. The images visualized are the states predicted by the learned world model. Each image is merged with an original image (left) and a reconstructed image (right). Each row evolves the state over time.

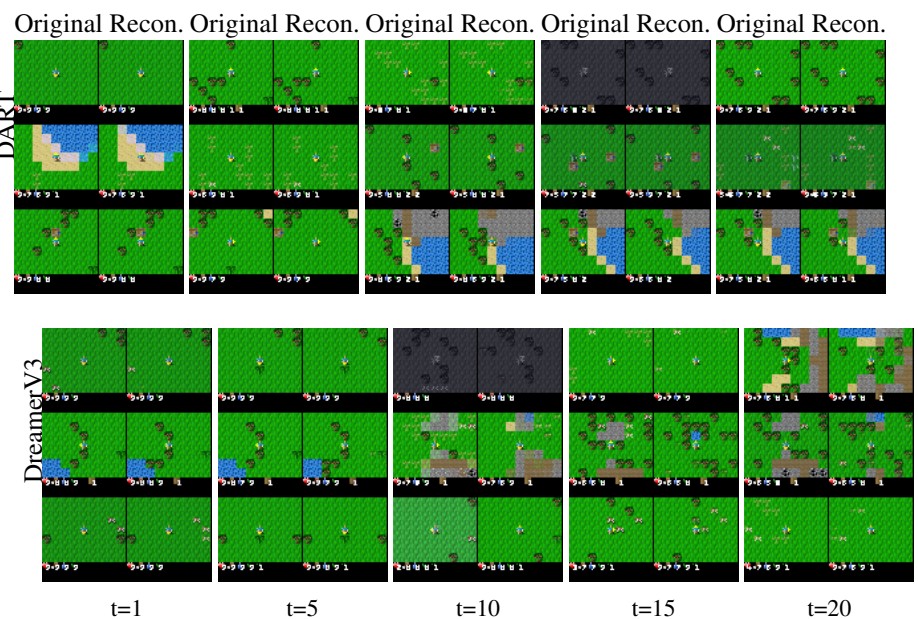

Figure 6: Three imagined trajectories for DART (top) and DreamerV3 (bottom) for Crafter environment with randomly sampled action sequences.

