# OpenReview forum: "Learning to Play Atari in a World of Tokens"
_ICLR.cc/2024/Conference — ICLR 2024 Conference Withdrawn Submission_

### Official Review · Reviewer_f8be · 2023-10-28

**Soundness:** 3 good
**Presentation:** 2 fair
**Contribution:** 2 fair
**Rating:** 6
**Confidence:** 3

**Summary:**

The paper proposes to use a transformer-based architecture for world modeling and policy learning. In addition, it uses VQ-VAE to obtain discrete representations of the observation and memory tokens to handle partial observability. Results on Atari 100k show the proposed method outperforms previous state-of-the-art methods that do not use look-ahead search.

**Strengths:**

* The paper proposes a transformer-based architecture for world modeling and policy learning and shows it's quite effective on Atari 100k.
* The paper conducts extensive experiments on Atari 100k and provides many metrics to demonstrate the superiority of the proposed method.
* The paper is easy to follow.

**Weaknesses:**

* From the results in Table 1, DART is worse than DreamerV3 on multiple games. Also, DreamerV3's results are missing from the figures.
* It would be better to evaluate DART's performance on multiple domains, such as robotic control, Crafter, DMLab, or even Minecraft, to show the discrete representations can generalize to different scenarios.
* It seems like the core contribution of the paper is from the architecture side. However, there are also many prior works that leverage the transformer architecture for world modeling. It would be good to clarify the novelties of this work and how the work differs from prior works. A system-level comparison table would be helpful.

**Questions:**

What's the reason for using an image-based VQ-VAE instead of a video-based VQ-VAE?

---

> ### Author Response · Authors · 2023-11-17
>
> We thank the reviewer for their time and feedback. Please find our responses below.
>
> >  From the results in Table 1, DART is worse than DreamerV3 on multiple games.
>
> While we agree that DART may not match DreamerV3’s performance in certain games, it’s essential to note their differences. DreamerV3 relies on a strong 18 million parameters model using CNN & GRU for Atari game training, whereas DART is a transformer-based world model with 3 million parameters. DART outperforms DreamerV3 in 11 out of 26 games, achieving similar superhuman scores. It excels in median score, iqm, and optimality gap scores compared to existing models. Moreover DART has better interpretability due to its use of self-attention, as illustrated in Figure 4. In a comparison with TWM, another transformer-based world model using a Dreamer-like objective function, DART appears to be more sample-efficient.
>
> > Also, DreamerV3's results are missing from the figures.
>
> Thanks for pointing this out. We have updated Fig. 2 and Fig. 3 with Dreamerv3 results in the revised version.
>
> > It would be better to evaluate DART's performance on multiple domains, such as robotic control, Crafter, DMLab, or even Minecraft, to show the discrete representations can generalize to different scenarios.
>
> While running on all the environments would be challenging during the discussion period due to the limited compute budget, along with the challenges of restructuring the model for continuous action space, we trained our model on the challenging Crafter benchmark. While the training is still ongoing preliminary results after 100k iterations indicate that DART surpasses IRIS and DreamerV3 on Crafter, and is more sample efficient as updated in Section A.1 (the final results for 1M iterations would be updated soon). It is important to note that Dreamerv3 uses 200 million parameter model for training Crafter while DART is limited to 3 million parameters only.
>
> >  It seems like the core contribution of the paper is from the architecture side. However, there are also many prior works that leverage the transformer architecture for world modeling. It would be good to clarify the novelties of this work and how the work differs from prior works. A system-level comparison table would be helpful.
>
> Through this work, we propose a MBRL approach which models the world and learns the behavior using discrete tokens, resulting in improved sample efficiency. This is done by using a ViT like architecture for learning the behavior by utilising discrete tokens. These discrete tokens are predicted by the world model using a GPT-like architecture. Moreover we propose a simplified approach for modeling memory again as discrete tokens using self-attention mechanism, further helping in learning the behavior. This is an improvement over IRIS which relies on CNN + LSTM for learning the behavior using  image reconstruction. We have included model configuration in Table 5 discussing the differences between the model design of recent MBRL approaches.
>
> > What's the reason for using an image-based VQ-VAE instead of a video-based VQ-VAE?
>
> Our primary aim is to simplify the existing approaches and model a policy using more abstract representations. The image-based VQ-VAE has shown improved image synthesis, by modeling precise low-level discrete elements. As a result, we chose to focus on the Image-based VQ-VAE and rely on the transformer for encoding temporal representation. While Video VQ-VAE could be a promising direction for future exploration, it is challenging to implement it within the short timeframe of rebuttal given our limited compute resources.
>
> We thank the reviewer again for constructive feedback. We hope we could address most of the reviewer’s concerns. We would appreciate it if they could kindly consider adjusting the score. We are happy to provide further clarification.

---

> > ### Author Response · Authors · 2023-11-21
> >
> > We kindly request the reviewer to please provide us with the final response, as the deadline for the discussion phase is fast approaching.

---

> > > ### Comment · Reviewer_f8be · 2023-11-21
> > >
> > > Thanks for the rebuttal! I increased my score to 6.

---

### Official Review · Reviewer_JiU2 · 2023-10-31

**Soundness:** 2 fair
**Presentation:** 2 fair
**Contribution:** 2 fair
**Rating:** 3
**Confidence:** 3

**Summary:**

This paper introduced discrete representation to transformer-based model-based RL. The world model is learned with a transformer-decoder. Unlike work, the policy is learned from the transformer-encoder, where self-attention can aggregate information. To handle situations where long-term dependency is required, the proposed method introduced a special memory token to pass information from a few steps ago. Experiments on Atari 100K showed improved median human-normalized scores.

**Strengths:**

- The paper is well-written and easy to follow.
- The ablation study provided in the paper is well-designed and informative.

**Weaknesses:**

- I need more clarification about the motivation that long-range dependencies impede Dreamers learning. Could the authors provide experiments comparing the world model accuracy among the proposed method and prior non-transformer-based methods on some long-range tasks?
- I think the world model accuracy should be measured more in detail (i.e. future states predicting accuracy. reward predicting accuracy, etc.) to fully support the author's arguments on the RNN-based world model and Transformer-based world model.
- Could the authors also compare the model capacity among DART and other baselines?
- Could the authors explain how the long-term dependencies are required in Atari tasks?
- Could the authors please provide the value set for some critical hyper-params? i.e. What is the value of $K$? How long is the Transformer horizon set?
- The MEM token compresses all past information into one vector, similar to the RNN hidden states. Could the authors explain why MEM helps?
- Could the authors also include the STORM[1] paper to compare and explain the advantages of DART over STORM?
- Could the authors add one ablation study on DART with continuous representation?

- [1] Weipu Zhang, Gang Wang, et al. STORM: Efficient Stochastic Transformer based World Models for Reinforcement Learning. Advances in Neural Information Processing Systems 2023.

**Questions:**

Please see weaknesses.

---

> ### Author Response · Authors · 2023-11-17
>
> We thank the reviewer for their time and feedback. Please find our responses below.
>
> > I need more clarification about the motivation that long-range dependencies impede Dreamers learning. Could the authors provide experiments comparing the world model accuracy among the proposed method and prior non-transformer-based methods on some long-range tasks?
>
> Dreamer use GRU which models temporal representation as compressed states leading to degradation for long-horizon task. Some env in Atari test for long-horizon, we compared the performance of transformer-based models with a CNN+GRU-based model on crafter. Since world model accuracy directly determines the final policy performance [1], Table 3 of the revised version show that transformer-based models (~ 3 million parameters) exhibit better performance compared to DreamerV3 (~200 million parameters for Crafter). We will report the final performance of all the models with 1M iterations in the final version of the paper as experiments are currently running.
>
> >  I think the world model accuracy should be measured more in detail (i.e. future states predicting accuracy. reward predicting accuracy, etc.) to fully support the author's arguments on the RNN-based world model and Transformer-based world model.
>
> Fig. 5 (revised version) illustrates the predicted future trajectories of the transformer-based world model. Here, we show evolution of future within the learned world model given sequence of actions. We also visualize the next state prediction accuracy, the next reward prediction accuracy for our world model. Fig. 6 compares the predicted trajectories of DART with DreamerV3 on crafter and reports higher PSNR for DART.
>
> > Could the authors also compare the model capacity among DART and other baselines?
>
> Table 5 includes information about the model capacity as well as detailed model configuration for recent MBRL approaches. DART has almost identical model parameters as IRIS, while maintaining high performance. Dreamerv3 uses an 18M parameter model for Atari, resulting in enhanced capacity compared to other models.
>
> > Could the authors explain how the long-term dependencies are required in Atari tasks?
>
> Not all Atari games in general require long-term dependencies, however, the use of a transformer is still beneficial as, unlike RNN which provides the compressed version of the past states, transformer can access full state information, resulting in the modeling of complex relationships for better accuracy.
>
> > Could the authors please provide the value set for some critical hyperparams?
>
> Thanks for pointing this out. A detailed list of all hyperparameters used for each module has been added in the revised version in Section A.4.
>
> > Could the authors explain why MEM helps?
>
> The MEM token captures temporal representation through the self-attention mechanism. Temporal representation is modeled simultaneously with spatial representation, eliminating need for extra modalities, in contrast to prior approaches that independently process spatial and temporal aspects using CNNs for spatial and RNNs for temporal aspects. This simple approach ensures that the model selectively propagates task-relevant features, as illustrated in Fig. 4. Here the agent dynamically adjusts its attention to the memory in response to the task’s demands, simplifying the training process.
>
> > Could the authors also include the STORM paper to compare and explain the advantages of DART over STORM?
>
> STORM was accepted at Neurips 2023 and became accessible on arxiv on 14th October which is after the ICLR submission deadline. It leverages Transformer for world modeling and uses a categorical variational autoencoder that adds stochastic noise to the latent space. It demonstrates a mean human performance of 126.7% with the same superhuman score of 9. In contrast, when considering the median human normalized score, DART demonstrates better performance. However, STORM stands out for its remarkable reduction in training time, achieved through the use of a single stochastic latent variable. In contrast, DART employs discrete tokens, allowing the model to concentrate on task-relevant tokens. In the updated version, Table 9 has been extended to include the scores of STORM.
>
> > Could the authors add one ablation study on DART with continuous representation?
>
> Evaluating DART with continuous representation will involve modifying the methodology for representation learning, world modeling, and behavior learning. This is challenging to implement and validate during the short discussions phase, due to computational resources and long training time but it is an interesting follow-up work.
>
> [1] "High-accuracy model-based reinforcement learning, a survey." Artificial Intelligence Review.
>
> We thank the reviewer again for their insightful comments. We hope we have addressed the concerns. Please let us know if we can provide more clarifications. Please consider increasing the score if you find the responses satisfactory.

---

> > ### Author Response · Authors · 2023-11-21
> >
> > We kindly request the reviewer to please provide us with the final response, as the deadline for the discussion phase is fast approaching.

---

> ### Comment · Reviewer_JiU2 · 2023-11-23
>
> Thank the authors for the response.
>
> **R1**: long-range dependencies impede Dreamers learning
>
> DreamerV3 performs quite well on Crafter. If compressed states lead to degradation for the long-horizon task, could the authors provide a comparison between DreamerV3 on Crafter with converged results, not limited to 100K interaction? Also, the memory token used in the proposed method is also a compressed vector; I think the argument and experiment results are not convincing to me.
>
> **R2**: model accuracy comparing with baselines
>
> Is Figure 5 & 6 the reconstruction or generation? The label in the figures indicates it is reconstruction. It's a little confusing to me. I thought the authors should provide a quantitative comparison result (as requested) with baseline models to demonstrate the proposed method learns a more accurate world model.
>
> **R3**: MEM is also compressed token, could the authors explain why MEM helps?
>
> ConvRNN can also capture both spatial and temporal relationships; why is MEM better?
>
> **R4**: ablation on continuous representation.
>
> I think this experiment is quit important to verify the effectiveness of the proposed discrete tokens.

---

> > ### Author Response · Authors · 2023-11-23
> >
> > We would like to thank the reviewer for their insightful comments,
> >
> > R1: Indeed mem token is a compressed vector, however unlike recurrent networks our use of self-attention are less prone to vanishing gradient problem enabling efficient modeling of long -horizon task [1]. Considering the autoregressive inference of DART (similar to IRIS), it is challenging to train it for 1 million interactions during the rebuttal period mainly due to the limited compute budget and time. We currently have results for training for 200K interactions which we added to appendix section A.1, please see revised version of the manuscript. We will add longer training for the camera ready version.
> >
> > R2: The current sota baseline models such as IRIS, DreamerV3, and TWM did not release pre-trained weights of the world model. For this reason, it is required to re-train all the models which is challenging again considering the short discussion phase. Hence we relied on comparing the state reconstruction quality of Dreamver V3 and DART, along with its PSNR values which is often used in literature[2] and the final policy performance (Table 3) for evaluating world model accuracy as done in the past work [5] .
> >
> > R3: Modeling MEM as discrete tokens enables utilizing self-attention for capturing the spatio-temporal relationships. The hierarchical structure of self-attention with its local multi-headed self-attention requires fewer parameters and implicitly performs convolution operations with a much larger receptive field [3]. This enables it to simultaneously model spatial-temporal relationships with fewer parameters enabling better performance for our proposed model.
> >
> > R4: We agree that modeling DART for continuous actions is important and can be an exciting future work. However, this requires significant change in our model structure. Moreover, most of the past MBRL models proposed for improving sample efficiency such as IRIS, TWM, and STORM (recommended by the reviewer) [4] perform comparisons only on Atari-100K.
> >
> > While we did our best to address all of the reviewer’s concerns, we would like to emphasise that basing the evaluation of our work on a comparison with contemporary work, or work which became available after the submission deadline [4] is against ICLR guideline (please refer to https://iclr.cc/Conferences/2024/ReviewerGuide FAQ section). It is also recommended to ask for experiments that do not significantly change current content but can be used to validate proposed method.
> >
> > We thank the reviewer again for their engagement and constructive feedback which helped to improve our manuscript. Given limited time and resources, we did our best to address the concerns and we would like to request that the reviewer adjust the score accordingly.
> >
> > [1] “When Do Transformers Shine in RL? Decoupling Memory from Credit Assignment. Neurips 2023”
> >
> > [2] "An energy efficient edgeai autoencoder accelerator for reinforcement learning. IEEE Open Journal of Circuits and Systems.”
> >
> > [3] "On the relationship between self-attention and convolutional layers. ICLR 2020."
> >
> > [4] Weipu Zhang, Gang Wang, Jian Sun, Yetian Yuan, Gao Huang. STORM: Efficient Stochastic Transformer based World Models for Reinforcement Learning. Advances in Neural Information Processing Systems 2023.
> >
> > [5] “High-accuracy model-based reinforcement learning, a survey. Artificial Intelligence Review.”

---

> > > ### Comment · Reviewer_JiU2 · 2023-11-23
> > >
> > > I think the experimental verifications (i.e., long-term dependency, world model accuracy) I ask for can be done much easier on some well-designed toy experiments and thus do not require many computational resources. The authors' statements without concrete experimental support are unacceptable to me.

---

> > > > ### Author Response · Authors · 2023-11-23
> > > >
> > > > Thank you for your quick response.
> > > >
> > > > We have addressed model accuracy in Appendix A.5 and long-term dependency in Appendix A.1.

---

### Official Review · Reviewer_vBaD · 2023-11-02

**Soundness:** 3 good
**Presentation:** 3 good
**Contribution:** 2 fair
**Rating:** 5
**Confidence:** 4

**Summary:**

This paper introduces a model-based RL method to learn to play atari in a sample-efficient manner (100k environment interactions). The basic approach of this paper is similar to various MBRL papers wherein they first learn a state representation followed by learning the dynamics of the environment which is followed by learning the policy in the imagined environment. In this paper, the above 3 steps are carried out as follows:

- Representation learning: The proposed method learn a discrete representation of the state space using a vq-gan style approach. Each frame is encoded to K tokens from a codebook.

- Dynamics learning: The proposed method utilize a transformer to learning the dynamics. The dynamics learning is comprised of 3 objectives - (1) predicting next token given the previous tokens, (2) predicting the reward for each frame, (3) predicting whether the episode has ended.

- Policy learning: The proposed method use actor-critic to learn the policy. The policy is parametrized as a ViT encoder which takes as input - (1) A cls token, (2) A set of codes from a particular frame, (3) A mem token which aggregates information from past frames. The output corresponding to the CLS token is used to output the action and the value. The output corresponding to the mem token is used as input the ViT for the next frame.

They show that their approach results in state-of-the art performance across 26 atari games when measuring median performance. Furthermore, they present various ablations such as visualizing the attention matrix of the policy and evaluating the model without various components such as positional embeddings, exploration, mem token etc.

**Strengths:**

- The main strength of this paper is in the clarity of the idea and the presentation. The paper combines various existing approaches and combines them in a way that is not too complicated to understand or implement.

- The novelty over the previous approach with discrete tokens - IRIS -  lies in being able to learn a policy on latent states rather than on reconstructed observations as done in IRIS. The advantage of not using reconstructed observations is that it is a lot more computationally efficient to use the latent states directly.

**Weaknesses:**

These are not weakness per se, but the reviewer thinks In these respects paper can be improved:

- The approach is simple (which is good) and integrates components used by the model already exist in literature. Learning a world models on discrete tokens has been previously introduced in IRIS and using a ViT policy (which the authors claim to be their main novelty) head has been studied by Yoon et al 2023 (https://arxiv.org/abs/2302.04419). Usage of a memory token to feed past context has also been studied in Bulatov et al 2022 (https://arxiv.org/abs/2207.06881), Didolkar et al 2022 (https://arxiv.org/abs/2205.14794), Moudgil et al 2021 (https://arxiv.org/abs/2110.14143). I would suggest to include these works in the introduction and recontextualize the work based on the above works.

- The results dont seem strong enough. The model only differs from IRIS in the policy and in many games IRIS still performs better. Secondly, according to the IRIS paper, they achieve superhuman performance in 10 games while in Table 1 it says 9 games and DART also achieves superhuman performance in 9 games. Therefore, IRIS actually can outperform humans in more games than DART hence the usefulness of of the ViT policy is not very apparent. It would be nice to see a section where the authors compare DART to only IRIS and try to study in more detail the importance of the ViT policy. Figure 3b compares DART to each approach individually but I am not sure how these probabilities are calculated. Can the authors clarify this?

[More of a comment than a weakness]. While the current paper and the baselines study the performance in a setting where the model is limited to 100k interactions, I think it would still be useful to compare how these approaches scale with more interactions and whether the current trends still hold with more interactions.

**Questions:**

See weakness section.

---

> ### Author Response · Authors · 2023-11-15
>
> We thank the reviewer for their insightful comments and valuable time.
> >The approach is simple (which is good) and integrates components used by the model already exist in literature. Learning a world models on discrete tokens has been previously introduced in IRIS and using a ViT policy (which the authors claim to be their main novelty) head has been studied by Yoon et al 2023 (https://arxiv.org/abs/2302.04419). Usage of a memory token to feed past context has also been studied in Bulatov et al 2022 (https://arxiv.org/abs/2207.06881), Didolkar et al 2022 (https://arxiv.org/abs/2205.14794), Moudgil et al 2021 (https://arxiv.org/abs/2110.14143). I would suggest to include these works in the introduction and recontextualize the work based on the above works.
>
> In the updated version, we have revised the introduction section to incorporate the suggested works. We want to highlight that Yoon et al’s work (https://arxiv.org/abs/2302.04419) involves pretraining a transformer encoder to model object relationships. This generates representation used for downstream RL, potentially encoding task-irrelevant information. However, our approach employs a transformer encoder for policy modeling without any pretraining, allowing the agent to learn the policy by focusing on task-relevant information. Bulatov et al 2022 (https://arxiv.org/abs/2207.06881) require storing all memory tokens from the past segment for prediction, however our method, employs a self-attention mechanism to propagate information as required by the task from the current time step to the next, eliminating the need to store any past tokens (refer to Fig. 4). Our model simplifies memory representation, in contrast to Didolkar et al’s  (https://arxiv.org/abs/2205.14794)  two-stream network which uses a slow stream (long-term memory) network modeled using a recurrent network and a fast stream (short-term memory) modeled using a transformer which is conditioned on the slow stream.
>
> > The results dont seem strong enough. The model only differs from IRIS in the policy and in many games IRIS still performs better.Secondly, according to the IRIS paper, they achieve superhuman performance in 10 games while in Table 1 it says 9 games and DART also achieves superhuman performance in 9 games. Therefore, IRIS actually can outperform humans in more games than DART hence the usefulness of of the ViT policy is not very apparent.
>
> DART has better performance than IRIS in 18 out of 26 games (underlined in the revised version in Table 3). From the Table 1 of original IRIS paper (https://openreview.net/pdf?id=vhFu1Acb0xb) and from Table A.1 of Schwarzer et. al (https://arxiv.org/pdf/2305.19452.pdf) IRIS outperforms humans in 9 environments only. We achieve the same superhuman score of 9, we show a significant improvement in other metrics including median, IQM, and optimality gap scores when compared to IRIS, along with better interpretability of the policy.
>
> > It would be nice to see a section where the authors compare DART to only IRIS and try to study in more detail the importance of the ViT policy.
>
> In the revised version, in Table 1 we try to highlight DART’s superior performance compared to IRIS. We also provide a system-level comparison of IRIS and DART in Section A.3. Our modeling of states and memory in the form of tokens allows ViT to enhance interpretability as shown in Figure 4.
>
> > Figure 3b compares DART to each approach individually but I am not sure how these probabilities are calculated. Can the authors clarify this?
>
> Introduced in [1] the metric used in Figure 3b is used to overcome the limitations of point estimates, as point estimates can vary across independent runs. We use this by collecting scores for 5 different seeds across 26 games for DART along with the model it needs to be compared with. Subsequently, the Mann-Whitney U-statistic is applied, averaging across all scores.
>
> > While the current paper and the baselines study the performance in a setting where the model is limited to 100k interactions, I think it would still be useful to compare how these approaches scale with more interactions and whether the current trends still hold with more interactions.
>
> While it is computationally expensive to run the model for millions of interactions, we are currently training our model and have initial results for DART on a few environments with 150k interactions (Section A.2). We are currently waiting for the corresponding results with IRIS and DreamerV3 and hope to be able to share them before the end of the discussion period.
>
> Given our limited compute resources, we hope we could address most of your concerns except the results for the longer IRIS experiment which we hope to provide soon. Kindly consider increasing the score if you find our responses satisfactory. We would be happy to answer any further questions.
>
> [1]. "Deep reinforcement learning at the edge of the statistical precipice." Advances in neural information processing systems 34 (2021).

---

> > ### Author Response · Authors · 2023-11-21
> >
> > We kindly request the reviewer to please provide us with the final response, as the deadline for the discussion phase is fast approaching.

---

### Author Response · Authors · 2023-11-22
**Request for Your Final Comments on the Revised Version**

Dear AC and Reviewers,

Thank you for your time and feedback on our paper. We've considered your comments and made improvements outlined in the rebuttal. As the discussion phase wraps up soon, we'd appreciate your final thoughts, considering our revisions.

Thanks,

Authors